# User Experiences with a Virtual Alcohol Prevention Simulation for Danish Adolescents

**DOI:** 10.3390/ijerph17196945

**Published:** 2020-09-23

**Authors:** Julie Dalgaard Guldager, Satayesh Lavasani Kjær, Patricia Lyk, Timo Dietrich, Sharyn Rundle-Thiele, Gunver Majgaard, Christiane Stock

**Affiliations:** 1Unit for Health Promotion Research, Department of Public Health, University of Southern Denmark, 6700 Esbjerg, Denmark; satakjaer@health.sdu.dk (S.L.K.); cstock@health.sdu.dk (C.S.); 2Research Department, University College South Denmark, 6100 Haderslev, Denmark; 3Game Development and Learning Technology, The Maersk Mc-Kinney Moller Institute, University of Southern Denmark, 5230 Odense, Denmark; pabl@mmmi.sdu.dk (P.L.); gum@mmmi.sdu.dk (G.M.); 4Social Marketing @ Griffith, Griffith Business School, Griffith University, Nathan QLD 4111 Brisbane, Australia; t.dietrich@griffith.edu.au (T.D.); s.rundle-thiele@griffith.edu.au (S.R.-T.); 5Institute for Health and Nursing Science, Charité—Universitätsmedizin Berlin, corporate member of Freie Universität Berlin, Humboldt-Universität zu Berlin, and Berlin Institute of Health, 13353 Berlin, Germany

**Keywords:** students, pupils, user experiences, virtual reality, alcohol prevention, drug resistance skills, school

## Abstract

This pilot study explores 31 Danish adolescent user experiences for the newly developed virtual party simulation app—Virtual Reality (VR) FestLab. The main objective of this study was to investigate usability for VR FestLab, which aims to improve alcohol resistance skills for Danish adolescents. A secondary objective was to understand gameplay experiences. The study is a mixed method study that draws on questionnaire data (*n* = 31) and focus group interviews (*n* = 10) of boarding school students participating in the pilot study. Descriptive statistics were used to examine quantitative data, and qualitative data were analyzed thematically. Quantitative findings indicated that gameplay experiences of the VR simulation were positive, and all User Experience Questionnaire (UEQ) items were answered positively. The focus group interviews showed that adolescents found the simulation to be realistic. Feedback indicated that the group pressure experienced in the simulation was regarded to be less than in real life. Adolescents had varying approaches to playing the VR simulation, they thought the quality of the simulation was good, and only a few users experienced technical difficulties. These initial study findings indicate that VR FestLab is a promising tool for the prevention of alcohol use among adolescents.

## 1. Introduction

Adolescent alcohol consumption is a major public health concern because of its short- and long-term psychological, social, and physical health consequences [1,2]. Heavy alcohol drinking during adolescence has been associated with cognitive deficits in learning, attention, and communication skills, disruptions in memory, increased susceptibility for anxiety, and increased risks of substance use disorders later in life [3]. Further, alcohol substance use has been found to be associated with delinquency, unwanted pregnancy, and school failure [4]. Temporal trends indicate a decrease in alcohol use among European adolescents from 1995 to 2018 [5,6,7]. However, it is worrying that compared to 45 countries and regions in Europe and Canada, Danish adolescents have the highest prevalence of drunkenness [6]. Moreover, the latest data from 2018 indicate a slight increase in alcohol use among Danish adolescents, where 26% of 15-year-old boys and 18% of 15-year-old girls drink alcohol at least once a week [8].

Considering this high prevalence [5,7], and the health and social consequences of alcohol use [1,2,3], it is crucial to continue to focus on the prevention of alcohol consumption in adolescents. One of the novel approaches used in alcohol prevention is the usage of new technologies, such as virtual reality (VR). Using VR has the advantage that complex information can be presented to the audience in an accessible way in a format which is engaging and easily understood [9]. The usage of VR technology in alcohol studies is still in its infancy [10], and in a review of the prevalence hereof, only one study had its focus on adolescents [11].

The majority of existing school-based prevention programs within this area include elements of refusal skills training and focus on resisting peer pressure [12]. Further, it is well-established that refusal self-efficacy plays an important role in the development and maintenance of drinking behavior (e.g., Young and Oei [13]). Using VR in alcohol prevention for adolescents is a good opportunity to turn the more traditional method of role play for drug resistance skills into an experimental VR environment. Previous anti-alcohol campaign efforts that focused on presentation of factual knowledge, scare campaigns, or moral encouragement have mostly been proven ineffective [14,15], suggesting alternate approaches are warranted. Approaches such as VR allow adolescents to experiment with (virtual) alcohol intake in a safe environment, excluding potential negative physical risks that occur when experimentation occurs in real-life settings.

Using VR technology in alcohol prevention remains innovative, with few examples evident [16]. Very few VR applications have been developed within this area, and limited understanding of user experiences is available [17]. Within the area of VR for learning in general, immersion and presence are important elements of the user experience [18,19]. Presence is often described as the feeling of being present in the virtual environment [20,21], and immersion as the physical properties that make the user feel present [22]. Immersion is constantly linked to VR and describes a complete, intense, and immersive virtual experience [23]. When users feel present in a virtual environment, they tend to respond realistically to the situation they experience [20], which is especially beneficial when designing experiential learning.

More generally, for interaction design, the goal is usability, which attempts to ensure that a product is *“easy to learn, effective to use, and enjoyable from the user’s perspective”* [24] (p. 14), and thus improves the user experience. This, for example, includes elements of design that ensure perceived and actual properties of the included features match [25]. Salzman [26] linked the user’s interaction experience and learning experience in VR directly to the learning outcome—poor experiences in an application can affect the user’s learning outcome in a negative way. Therefore, as a designer, it is necessary to examine user experiences through development in iterative cycles of design, test, and evaluation [24].

The purpose of this pilot study was to explore the user experiences of VR FestLab (University of Southern Denmark, Odense, Denmark)—a newly developed Danish virtual party simulation, aimed at improving alcohol resistance skills among adolescents. The objective was to investigate participants’ feedback towards the usability of VR FestLab. A secondary objective of this study was to investigate the adolescents’ gameplay experiences of VR FestLab.

## 2. Materials and Methods

### 2.1. The VR FestLab Prototype

VR FestLab is a virtual reality computer simulation of a three-dimensional typical party situation for adolescents. When the player enters the simulation, the user can steer through the party through a series of realistic choices presented to the player. The player can approach different people at the party and across different interactions, they are confronted with different behavioral choices. Drinking choices involve peers asking players if they would like an alcoholic or non-alcoholic drink. Other gameplay options include choices to: play a drinking game; support peers who are drunk; dance with their peers, and more (see Figure 1).

For more information on details of VR FestLab, see Lyk et al. [27], and for details outlining the development process applied to the design of VR FestLab, see Vallentin-Holbech et al. [28].

### 2.2. Participants

The current pilot study used a purposive sample of students from a Danish boarding school. This was chosen to ensure good geographical coverage of study participants, since boarding school students in general have their home address and socio-cultural background in different Danish regions. The study used a mixed method approach [29] and adheres to the understanding that a mixed method study incorporates both qualitative and quantitative methods at data collection and analysis within a single study [30]. Students were asked to try out VR FestLab individually. Immediately thereafter, they filled out a standardized questionnaire, followed by their participation in a focus group interview. Ten focus group interviews were conducted (*n* = 31, average age 16, SD ± 0.71). Focus groups were chosen as opposed to individual interviews, for several reasons. Firstly, because of the age of the students, they were interviewed together with their peers to provide confidence. Second, focus groups were chosen to overcome social desirability bias, giving students the opportunity to express negative views on VR FestLab. Smaller focus groups of 3-4 students per group were chosen to capture individualized feedback, allowing different perspectives and opinions for each adolescent to emerge while avoiding group pressure.

### 2.3. Data Collection

This pilot study is based on questionnaire data from the students from the boarding school, as well as on semi-structured focus group interviews with open-ended questions with the same group of students. Themes for the interviews focused on the students’ opinions about VR FestLab, their attitudes towards working with alcohol in this way, and their engagement in VR FestLab.

Interviews ranged from 13 to 47 min in length (mean 32 min, SD ± 9.90). Group interviews were digitally recorded, and students’ names were replaced by pseudonyms. Interviews were transcribed verbatim by two research team members, resulting in a total of 124 pages. To facilitate reporting in English, this research provides quotes translated from the original data in Danish. Interviews were analyzed with the help of the data analysis software program NVivo 12 (QSR International Pty Ltd., Doncaster, Victoria, Australia). A researcher from the research team, who was not involved in the co-creation development process of the VR FestLab, analyzed and coded all the material using the six phases of thematic analysis recommended by Braun and Clarke [31]. First, the researcher got intimately familiar with the raw data, both during and after the transcription phase, which made it possible to code every data item that represented obvious, as well as underlying meanings. These codes were, in most cases, coded again in the process of searching for themes, where coherency and meaningful patterns were identified. These themes were thereby reviewed and discussed together with another researcher from the team to establish that a clear definition of each relevant theme was made alongside with writing the analysis. Finally, consensus of other members from the research team was used to develop the final structure of the relationship between the themes and selected representative data [31].

Quantitative data were obtained from a standardized self-administered paper questionnaire (*n* = 31). Demographic information contained data on age and sex. As a proxy for family socioeconomic status, we used the Perceived Family Affluence from the Health Behavior in School-aged Children study, asking how well-off the adolescents found their family to be [32]. Alcohol use, been drunk, and binge drinking were measured by items developed by the research team, asking if students had ever been drinking alcohol, been drunk, or drunk five or more drinks at the same occasion. Lifetime alcohol use, been drunk, and binge drinking was measured using three items from the Danish youth survey “The monitoring of young people’s lifestyle and daily life” (MULD) [33], asking the students on how many occasions within the last 30 days they had been drinking alcohol, been drunk, or drunk five or more drinks at the same occasion. Questions regarding game enjoyment consisted of seven questions. Users were asked to report on the degree that students agreed that they liked to try the game, would like to explore the game further, did not like the VR experience, would recommend the game to their friends, thought the characters in the game were unrealistic, thought the game was realistic, or did not like the music. Answers to negatively worded questions were reversed for analysis (5-point Likert scale from “disagree a lot” to “agree a lot”). Gameplay experiences were measured using the short version of the User Experience Questionnaire (Short UEQ), with the eight parameters of the game being: obstructive/supportive, complicated/easy, inefficient/efficient, confusing/clear, boring/exciting, not interesting/interesting, conventional/inventive, and usual/leading edge [34]. Percentages were calculated to describe the characteristics of the students. Analyses were conducted with IBM-SPSS for Windows software, version 25.

The study adheres to Danish standards for ethical conduct of scientific studies and was approved by the Research Ethics Committee of the University of Southern Denmark on 18 December 2019 (case no 19/66794). The purpose of the study was explained to the students and they were informed that data would be presented in a completely anonymized form. In accordance with the Declaration of Helsinki, all students were informed that their participation was voluntary, and participants gave their oral consent. Parents gave written consent for their children’s participation.

## 3. Results

The results of the quantitative data will first be presented below. Thereafter, the qualitative results will be presented, and where relevant for the analysis, the quantitative results will be integrated.

### 3.1. Quantitative Results

The students were, on average, 16 years old (SD ± 0.71) and were evenly distributed regarding gender (55% female). The majority of the students were either very, or somewhat experienced in playing virtual reality games; some were a little experienced, and only one did not have any VR experience at all. Below, Table 1 presents the characteristics of the respondents regarding perceived family affluence and alcohol usage.

Figure 2 presents the results of the seven aspects of game enjoyment of VR FestLab. Results indicate that students were generally very positive regarding game enjoyment. The one item students were less enthusiastic about was the music of the game, where most students responded “don’t know” or felt that there was room for improvement.

Figure 3 presents the results of the eight components of the User Experience Questionnaire (Short UEQ) (7-point Likert scale from −3 (horribly bad) to +3 (extremely good)). All items received a positive mean (between 0.7 and 2.1), with only two items (obstructive/supportive and inefficient/efficient) being evaluated as neutral (mean between −0.8 and 0.8) and the rest being positively evaluated.

### 3.2. Qualitative Results

Two overarching themes, namely: *“Attitudes towards VR FestLab as an alcohol prevention tool”* and *“What can adolescents learn from VR FestLab?”* emerged. Participants’ feedback about the usability and gameplay experiences of VR FestLab can be characterized by three overarching themes: *“Adolescents’ perceptions of VR FestLab”, “VR FestLab versus a real life party”,* and *“VR FestLab gameplay experiences”*. These themes are presented next.

#### 3.2.1. Attitudes towards VR FestLab as an Alcohol Prevention Tool

Students were asked how they felt about working with issues regarding alcohol use among adolescents through a virtual reality simulation like VR FestLab. Most students were very positive towards a virtual reality experience like this, and the most frequently mentioned positive topic was that the users gained insights into experiencing a real party, like William, who explained:


*I think it gives a quite good image of a party, and I think it’s a really good way to teach people what a party is like without having to attend one [in real life].*


The analysis revealed that students experienced with drinking alcohol, like William above, believed that the party situation depicted in VR FestLab was very realistic. Students who were inexperienced with alcohol expressed that they enjoyed being introduced to how such a party could be. It should be mentioned though, that two girls explained that since VR FestLab is perceived as a game, they believed that people would not take VR FestLab seriously. They were worried that people would think it is just a fun party situation and would not consider the underlying learning experience about alcohol. This issue was not raised by other participants. In fact, below, Mia, a non-drinker, explains her opinion about the simulation:


*I have always learned that you should say no to drinking alcohol. VR FestLab is probably the best so far compared to what I have been learning from teachers and classmates, because the game allows you to make your own decisions and to see what happens if you to choose to drink alcohol or not. It is a quite cool approach, instead of just looking at a piece of paper, or something like that.*


Thus, Mia pinpoints how this approach is a superior alternative to the ways she has been exposed to alcohol education before.

Finally, when students were asked if they would recommend VR FestLab to others, they gave very different responses. Some were indecisive, and a few said no. However, the majority explained that it is not a simulation they would recommend if a friend was interested in playing a cool VR game, but they would recommend it if it was in connection to a talk about adolescents and alcohol, or if alcohol consumption would be discussed in a school setting.

#### 3.2.2. What can Adolescents Learn from VR FestLab?

Students were not coherent in the answers they gave as to what they could learn from the simulation in general. This could be due to VR FestLab permitting very different gameplay experiences depending on which decisions one chooses in the simulation. A few students expressed that they did not understand the idea of VR FestLab, while other students mentioned that they could see how the simulation can increase understanding about alcohol, its effects, and that they could learn how to say no to drinking in a party situation, like Charlie:


*I feel that it taught me [about alcohol] from a slightly different perspective. For those who have a hard time saying no [when offered a drink], they may also need to try the game and find out that it’s actually okay to say no to a drink.*


Other aspects which students mentioned that they had learned for the simulation were: How much alcohol you can drink before you pass out; how to handle drinking alcohol; how one could react if someone feels sick at a party; how to behave in different party situations in general; how the atmosphere is at parties; and how you can get affected by alcohol.

One of the tools used in the simulation to illustrate alcohol effects is via a graph featured like a classic game health bar, which illustrates the player´s blood alcohol concentration (BAC) at all times (see Figure 1). The BAC in the simulation increases as players accept alcohol, but decreases again over time to mirror the process that occurs in real life. Many students were very positive about the BAC bar, and below, Jennifer illustrates its relevance for both drinkers and non-drinkers and how you can learn something from it:


*I think it is applicable for both people who have tried drinking alcohol before, and for non-drinkers. It is also an important detail, that they can see when the alcohol gets out of your system.*


It should be mentioned that some students misunderstood the BAC bar and some found it to be unrealistic, and thereby they did not learn anything from it. However, many of the students who were very positive about the BAC bar expressed that their actions in the simulation were influenced by their BAC. For example, when asked whether his decisions were influenced by the BAC bar, Daniel explained that because of the BAC bar, he hesitated to drink more when the BAC bar was moving towards the highest end.


*My thoughts were like “no, maybe I shouldn’t just have a beer right now, maybe I should wait a bit so I’m a little sober”—and then I took a beer a little later...[...]...Yes, it was because of the BAC bar and I didn’t want to cross the limit, because then I knew I would black out.*


This shows how the BAC bar influenced the students’ choices. Whereas a few students experimented with trying to increase the BAC by often saying yes to alcohol, the majority said no to alcohol if the BAC bar was almost full.

Finally, differences in the perception of the simulation learnings in general between non-drinkers and alcohol drinkers were evident. Some of the experienced students mentioned that the simulation could probably not teach them so much, and that it was more seen as *just a game*, which was not mentioned by non-drinkers. However, students who were more experienced with alcohol were able to imagine how the simulation could be useful for non-drinkers, in that VR FestLab could prepare them for what a party with alcohol looks like.

#### 3.2.3. Adolescents´ Perceptions of VR FestLab

As presented earlier, most students (93.5%) liked the VR experience and all students enjoyed trying the simulation (Figure 2). The qualitative data reveal which aspects about the simulation the students liked. As an example, William explains here why he really enjoyed VR FestLab:


*I was pleasantly surprised and thought it was a pretty cool experience. I think the game was very well-constructed. And there were some cool scenarios to choose from.*


The simulation received very positive ratings in the Short UEQ questionnaire (Figure 3). In the interviews, when describing their positive experiences with the simulation, students mentioned words like *“cool and creative, catchy, with interesting scenes, exciting, fun, realistic, and with variation”*. Further, students found the simulation to be easy: *“…quite fun because it’s so easy. There are no instructions. It’s very easy to use“*, which was also supported by the quantitative results, where “ease” was the game experience most positively evaluated of the eight components of the Short UEQ questionnaire.

Although the students liked the overall variation of the scenes in VR FestLab, some mentioned that they experienced a little repetition in the simulation, and that the response options, when asked if they would like something to drink, were too similar. Like Jacob who explained:


*... if you chose to drink a beer with the guy who was a bit of a player, then he just kept asking, “you should have just one more beer”, “you should have just one more beer”, “you should have just one more beer”. That was all he could say.*


Further, students believed that the simulation was most interesting if you sometimes said yes to alcohol. A few perceived the simulation to actually encourage drinking alcohol and found the simulation to be a bit boring if you continuously said no to alcohol. Olivia explained:


*If you chose to drink a soft drink, there wasn’t much else to do in the game. The wild things would probably happen if you accepted alcohol. So, there should be something for us who says no, because otherwise it ruins the game a bit...*


Finally, from the analysis it became apparent that the scenes which were enjoyed most by the students were scenes which contained a lot of interaction between the game player and the characters. Examples are the scenes where the game player played different drinking games with interaction, and a scene where the game player has different options of how to help a friend who is feeling sick after having been drinking too much.

#### 3.2.4. VR FestLab versus a Real-Life Party

Most students (80.7%) thought the simulation was realistic and that the characters were realistic (74.2%) (Figure 2). William explains here:


*I think it was a surprisingly good example of a party. At least the ones I’ve experienced. I think you experience a lot of party atmosphere. But you didn’t notice it was a video at all. It actually seemed like you were there in reality.*


Some students even mentioned that they had felt awkward in some of the scenes because it was real people “standing” in front of them. This indicates further that the simulation was perceived as quite realistic. Another student, Alicia, added:


*I think it affected me because you could see the different things happening all around, and you were just like “inside” the game. You were also allowed to experience it yourself, and to make your mistakes in some way. Just inside a game.*


Ergo, Alicia pinpointed how it was important that you could make some mistakes in this safe environment. Students also mentioned that it was obvious that it was just a game and not an actual party and they found some specific features in the simulation to be less realistic. One example of this is the blood alcohol concentration bar. The BAC bar was developed so the BAC is calculated from the gameplayer´s chosen gender, the average weight for a 16-year-old female/male, and the game time used [24]. Several students mentioned that this BAC bar was unrealistic, in that it moved upwards too fast compared to how much you “drank” in the simulation.

In VR FestLab, you are presented with several situations of group pressure to drink alcohol. Many students explained how they felt that the group pressure experienced in VR FestLab was less than what they might experience in real life. For example, Eric explained:


*In real life, if someone says, “do you want a drink?” and you say no, then you can get a reaction where the other person does a face expressing: “okay you are such a bummer!”. Not that the other person has to say it, but you can feel it. Because the other person can give you a reaction like “Okay a boring type, okay a party pooper”. And you just don’t get that reaction [in the game] because you just walk away from the characters.*


#### 3.2.5. VR FestLab Gameplay Experiences

Students had very different approaches as to how they played the simulation. Many students were most interested in making choices so they could explore as many scenes as possible in the simulation. Some students made choices realistically, as they would have done in the real world; and then again, some used the simulation to experiment with being braver than they normally are. Many students played the simulation more than once, but had very different strategies of how to play the simulation again. Below, Susan explained her game strategy:


*If you want to experience it all, then you have to try it at least once more, if not twice, in order to try out the choices you want to. To try it all and see how it develops better if you drink more or less.*


Some, like Susan, were interested in going back to the same scene again, to explore the simulation further and see what would happen if you chose a different response option. Others played the first round like they would have acted in real life, and thereafter played the simulation again, this time experimenting to see what would happen if they said yes to a lot of alcohol, and still others chose the opposite strategy.

The majority of the students thought the quality of the simulation was good, where a few, like Mary, pinpointed that naturally, the simulation cannot be compared to commercial VR game productions:


*The game was good. Well, it could be a lot better, but considering that there are a lot of students participating, you probably didn’t have the best opportunities. But it wasn’t something you noticed—it wasn’t something I thought about so much.*


Despite a few students mentioning that the quality could be better, their gameplay experience was viewed positively overall.

Some students experienced some technical difficulties using VR FestLab. The most prominent technical challenge was that a few found some scenes to be blurred. Since most students did not have this experience, it is anticipated that this was due to the Oculus Quest device being used incorrectly by some users. Further, when waiting to choose your answer to a question in the simulation, the background of the simulation intentionally freezes, which some students found strange. Finally, in a scene where game players are on a dance floor, a few students mentioned that the music was too loud, which may explain the reason behind why only 26% of the students liked the music in general (Figure 2).

To sum up, the participants´ attitudes towards VR FestLab as an alcohol prevention tool was very positive, and they found the simulation to be realistic. Gameplay experiences were predominantly positive, but the group pressure experienced in VR FestLab was believed to be less than what the adolescents might experience in real life for some students. The participants had very different approaches as to how they played the simulation, and they thought the quality of the simulation was good, and only a few experienced some technical difficulties.

## 4. Discussion

In contrast to the more traditional methods of constructing alcohol prevention programs, VR FestLab demonstrates a novel approach that utilizes a virtual reality simulation, allowing participants to experience social pressure to drink while sober. Application of VR applications for the prevention of alcohol consumption remains in its infancy, with limited understanding of user experiences available [35]. Understanding the user experience is an important evaluation component, given the capacity to identify factors contributing to success or failure [16]. This mixed method research study evaluated user experiences for VR FestLab using the User Experience Questionnaire (Short UEQ), other process evaluation measures, and a group exit interview to gain adolescent user views. This pilot study delivers insights from participants on usability and gameplay experiences for the VR FestLab application.

Through questionnaire data and thematic analysis of focus group interviews with adolescents who had experienced VR FestLab, we found that overall, the adolescents experienced VR FestLab very positively. This is important for the learning outcome of VR FestLab, since it has been identified that the users´ interaction and learning experiences of virtual reality is directly linked to the learning outcome [26]. Our results of the Short UEQ are above the averages reported in other studies evaluating the usability of business software, web pages, web shops, and social networks [36].

The adolescents expressed that the overall learning output derived from the simulation was enhanced understanding of the different aspects about alcohol drinking and party behavior in general. Regarding the gameplay experience, adolescents found the simulation to be easy to use and navigate, and described the features of the simulation very positively (e.g., the simulation was cool and creative, catchy, with interesting scenes). Further, scenes that were enjoyed the most were scenes which contained a lot of interaction between the game player and the game characters. This result is in line with the research by Hudson et al. [37] who found that social interactions had a positive effect on the satisfaction of their underwater VR experience.

VR FestLab was perceived as a place allowing participants to make positive and negative choices in a safe environment. Adolescents found the simulation to be realistic, and some even felt embarrassed in some scenes, which indicated the participants’ immersion with the virtual experience. The term “immersion” is constantly linked to virtual reality [18,19] and is described as the subjective impression that you are participating in a comprehensive and realistic experience [38]. This can prove to be important for the learning outcome of the simulation, as immersion in VR has been found to enhance the educational output in terms of enabling multiple perspectives, situated learning, and transfer [39]. Further, that adolescents found VR FestLab to be realistic is consistent with the evaluation of an Australian VR alcohol simulation “VR House Party”, where realism was the strongest and most important feature raised by adolescents [16].

The majority of the adolescents found the quality of VR FestLab to be good, which is in contrast with the findings of the Australian VR alcohol simulation [16], where participants felt that the simulation quality required improvement. However, this difference was expected, since VR FestLab drew on more advanced technologies, and the development of the simulation was based on the experiences and lessons learned from the Australian game designers.

Our study provided important learnings which will inform future improvement. First, a few adolescents expressed that they did not understand the idea of VR FestLab, indicating that the simulation purpose may need to be revisited or more clearly articulated and/or integrated. Second, some adolescents believed that the simulation was more interesting if one said yes to drinking alcohol, rather than staying sober through the experience. This is an important learning, indicating that non-drinking experiences must be at least equally, if not more exciting than scenarios where players are saying yes to drinks. Third, results revealed that VR FestLab did not only present the theme of alcohol for the adolescents, but also brought up various feelings and thoughts about the party situation and socializing in general. Further, the majority of the adolescents highlighted that they would recommend VR FestLab as a component in a school prevention program. Thus, a further avenue of alike interventions could be, to a larger extent, to include a focus on social contact in general, or to integrate such a VR party simulation in broader school-based prevention programs, counselling, or therapy which has a focus on alcohol use, social contact, and group pressure. Fourth, some users felt that the group pressure experienced in VR FestLab was less than what they might experience in real life, which is an important issue, given this is one of the key aims of VR FestLab. Increasing the group pressure scenarios using more emotional cues when refusing drinks may be a way to achieve that. Fifth, a few adolescents found the Blood Alcohol Concentration (BAC) bar to be unrealistic, in that it was believed to rise too fast compared to how much you “drank” in the simulation. It is unknown if the students’ opinions regarding the BAC bar reflect an optimistic bias of the students of how much they can drink before they get drunk. Future iterations of the VR FestLab could feature the opportunity for players to select their gender and weight to improve accuracy of the BAC bar. The current algorithm was based on an average weight of a 15- to 18-year-old boy or girl and may not fit to individual body masses for game participants. Finally, a few experienced some technical difficulties when playing the simulation (e.g., blurred scenes). However, since the majority did not express such difficulties, we assume that this might be caused by the setup of the Oculus Quest device and not the simulation itself.

Cybersickness is an often-found unintended psychophysiological side effect of participation in a virtual environment, which can obstruct the user acceptance and have health and safety implications [40]. We found that the adolescents did not experience any adverse effects, such as cybersickness, when trying VR FestLab. Similarly, only five percent of participants in the Australian VR House Party study experienced cyber sickness [16], which is very low compared to other studies [40]. It should be noted that although we did not find any unintended side effects of VR FestLab in short-term use in a small sample, such side effects are known from other VR gaming [41]. To overcome any unintended side effects of VR FestLab, it was clearly stated in writing (before testing) and verbally (during testing) that the user should discontinue use if any vertigo, myopia, visual fatigue, or other problems should occur.

This study is not without limitations, upon which the results should be viewed, and which offer opportunities for future research. Firstly, due to the small sample in this pilot study, and the fact that the sample is based on adolescents solely from one boarding school, results should be interpreted with caution and may have been different with interviewees not in school or attending other types of schools. Further, the participating adolescents attended the same boarding school as the adolescents who, in the preceding school year, performed as actors in the VR FestLab simulation. As a result of this, some respondents mentioned that they had seen the acting persons before. This could have influenced the respondents’ perceptions of the VR FestLab simulation, steering their focus towards the performance of the actors and not the simulation per se. However, whenever this issue came up in the interview situations, respondents were instructed to set aside their knowledge of the acting persons and focus their answers to the questions about the simulation itself and not the actors involved.

Further, the adolescents may have been overly positive due to social desirability tendencies in the interview situation, and as a result, they could have reported more positive experiences with the VR FestLab. Although the groups were kept small to avoid group pressure, there is a remaining risk that some students may have had difficulties in forming an independent opinion in the interview session due to an opposing dominant attitude in the group. However, adolescents were informed that it was important for the study that they spoke openly and were assured strict confidentiality, which may have limited this type of bias.

It cannot be disregarded that the adolescents who volunteered to try the VR FestLab and to participate in the interview study may have felt more positive towards VR in general and may have been more experienced using VR games than the ones who did not volunteer to participate. Consequently, the adolescents may have been more positive towards VR FestLab. However, they may also have had higher expectations of the simulation, resulting in a more critical view of the simulation.

Finally, the results of this study will be used for the further development and implementation of VR FestLab; specifically, how the simulation can be integrated into school prevention programs. All data in this study were self-reported, and understanding of gameplay user experiences could be extended by including gameplay data. This would be an interesting avenue for future research to expand our understanding. Further, in the present study, the differences in game experiences between drinkers and non-drinkers and between adolescents with short-term and long-term alcohol use was not explored due to the small sample size; thus, it was not possible to assess group differences. This delivers a strong opportunity for future research. Moreover, more research is needed to establish long-term effects. Future research could build upon our results and explore any possible influences of sociodemographic characteristics (such as gender and ethnicity) of the users on their user experiences of VR prevention tools, since research in this area remains limited.

## 5. Conclusions

VR FestLab is one of the first virtual alcohol prevention simulations that have been developed and is focused on improving alcohol resistance skills in adolescents. This pilot study involving Danish adolescents demonstrated that VR FestLab was very positively evaluated by its target audience regarding its usability and game play experiences. Finding this positive user interaction with VR FestLab is an important aspect of the overall preventive impact of the simulation, even though the effectiveness of the learning outcome (improving alcohol resistance skills) is still to be tested in a future randomized controlled trial.

## Figures and Tables

**Figure 1 ijerph-17-06945-f001:**
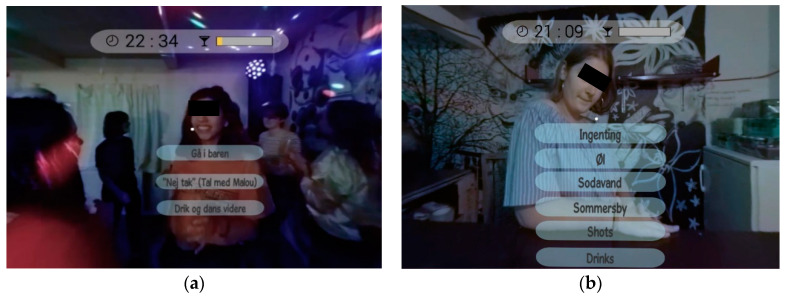
Screenshots illustrating two of the decision points in VR FestLab. (**a**) Scene: Dancefloor. Girl asks: “Would you like something to drink?” Answer possibilities: “Go to the bar”, “No thanks (talk to Malou)”, “Drink and keep on dancing”. (**b**) Scene: Bar. Bartender asks: “Would you like something to drink?” Answer possibilities: “Nothing”, “Beer”, “Soft drink”, “Breezer”, “Shots”.

**Figure 2 ijerph-17-06945-f002:**
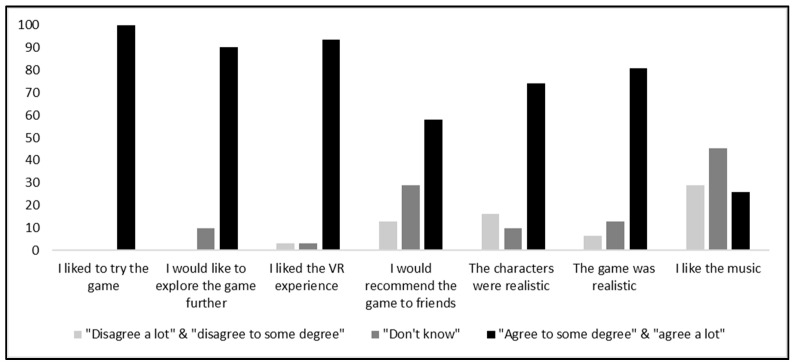
Game enjoyment of students in percentages (*n* = 31).

**Figure 3 ijerph-17-06945-f003:**
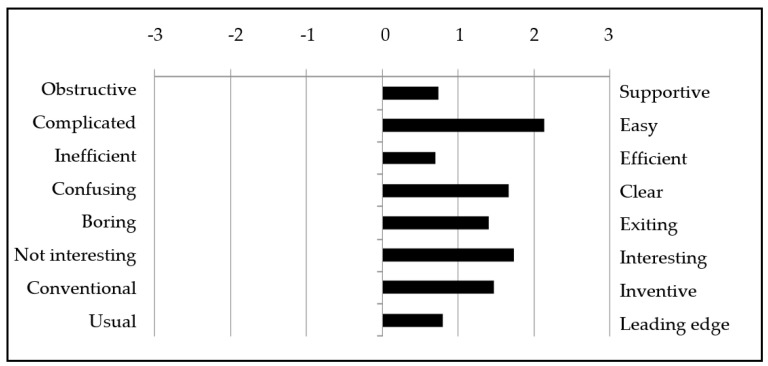
Game experiences (Short User Experience Questionnaire (UEQ)) of students, mean value (*n* = 31).

**Table 1 ijerph-17-06945-t001:** Characteristics of the respondents (*n* = 31).

Characteristics	Total	
Perceived family affluence, *n* (%)		
Quite well-off ^a^	7	(22.6)
Average	21	(67.7)
Not well-off ^b^	3	(9.7)
Lifetime alcohol use (at least one drink), *n* (%)		
No	3	(9.7)
Yes	28	(90.3)
Lifetime ever been drunk, *n* (%)		
No	7	(22.6)
Yes	24	(77.4)
Lifetime binge drinking, *n* (%)		
No	7	(22.6)
Yes	24	(77.4)
Alcohol use within the last 30 days, *n* (%)		
Never	11	(35.5)
One time	6	(19.4)
Two or more times	14	(45.2)
Been drunk within the last 30 days, *n* (%)		
Never	17	(54.8)
One time	10	(32.3)
Two or more times	4	(12.8)
Binge drinking within the last 30 days, *n* (%)		
Never	17	(54.8)
One time	7	(22.6)
Two or more times	7	(22.6)

^a^ No despondences in the category of “Very well-off”. ^b^ Response options “Not so well-off” and “Not at all well-off” combined.

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
