# Peer review of "User Experiences with a Virtual Alcohol Prevention Simulation for Danish Adolescents"

_ijerph, 2020, doi:10.3390/ijerph17196945_

Round 1

Reviewer 1 Report

A pilot study for a virtual party simulation app - VR FestLab - is reviewed in the manuscript User experiences with a virtual alcohol prevention simulation for Danish adolescents. The aim of the app is to improve alcohol resistance for adolescents.

The subject matter, the study and the line of thought is clearly presented in the manuscript where an interesting, innovative and important contribution to alcohol prevention for adolescents is introduced. However, I have a few suggestions for improvement, as follows:

Lines 108-109

I suggest deleting “please” in this sentence:

For more information on details of VR FestLab, please see Lyk et al. [27] and for details outlining 108 the development process applied to the design of VR FestLab, please see Vallentin-Holbech et al. [28].

Lines 119-121

The meaning of the following sentence is unclear and the word “students” comes twice.

Focus groups were chosen as opposed to individual interviews, for several reasons. Firstly, because of the age of the students, students were interviewed together with their peers to provide confidence.

I also suggest that individual interviews, rather than the group interviews, may have been more appropriate because of group pressure within the focus groups, where some of the adolescents may have had difficulties in forming an independent opinion because of dominant attitudes of other group members.

Lines 122-124

Use “pressure” instead of “think”.

Smaller focus groups of 3-4 students per group were chosen to capture individualized feedback allowing different perspectives and opinions for each adolescent to emerge avoiding group think pressure.

Lines 133-136

However, the majority explained that it is not a simulation they would recommend if a friend was interested in playing a cool VR game, but they would recommend it if it was in connection to a talk about adolescents and alcohol, or if alcohol consumption would be discussed in a school setting.

The above point of view is important to the subject matter of the article and relates to how the study can contribute to further development and application of the App VR FestLab.

The discussion with the adolescents afterwards also indicated that the app brought up various feelings and thoughts for the participants about partying and socializing, in addition to the alcohol use.

A further avenue that the app and a following interview process could provide is to also focus to a larger extent on social contact, group pressure and partying, while another one is to integrate he game as one component into settings like school prevention programmes, counseling and therapy where the subject of social contact, group pressure and alcohol use is discussed and worked with.

There were interesting points brought up in the focus groups that would be possible to follow up for further discussion with other game players, such as:

Lines 313-315

If you chose to drink a soft drink, there wasn't much else to do in the game. The wild things would probably happen if you accepted alcohol. So, there should be something for us who says no, because otherwise it ruins the game a bit...

Lines 350-354

In real life, if someone says, "do you want a drink?" and you say no, then you can get a reaction where the other person does a face expressing: "okay you are such a bummer!". Not that the other person has to say it, but you can feel it. Because the other person can give you a reaction like "Okay a boring type, okay a party pooper". And you just don't get that reaction [in the game] because you just walk away from the characters.

I’m aware that integrating the game into school prevention programs, therapeutic situations and counseling is a subject that is a further avenue for the program, but nevertheless it may be mentioned in the manuscript as a further development derived from the data provided by the study.

Reviewer 2 Report

Review report on

User experiences with a virtual alcohol prevention simulation for Danish adolescents

Summary

Nowadays, the application scenarios of virtual reality technology are becoming more and more extensive. For the prevention of alcohol use among Danish adolescents, this study introduced a newly developed virtual reality (VR) approach to improve the adolescents’ alcohol resistance ability, and in the meantime, to explore the user experiences and to investigate the usability for the VR FestLab used in the study. Descriptive statistics were used to examine quantitative data and qualitative data, and finally, these study indicate that the VR FestLab is a promising tool for the prevention of alcohol use among Danish adolescents.

General comments

As mentioned in this study, adolescent alcohol consumption is a major health concern because of its psychological and physical health consequences. Heavy alcohol drinking during adolescence has been associated with cognitive deficits in learning, attention and communication skills, disruptions in memory, increased susceptibility for anxiety later in life.

Considering the high prevalence of Danish adolescents’ alcohol use, it is crucial to focus on the prevention of alcohol consumption in adolescence. Compared with the other traditional methods, one of the novel approaches used in alcohol prevention is the usage of new technologies such as virtual reality. By using VR technologies, we can make the complex information presented to the audience in an accessible way which is engaging and easily understood. So this study introduced the VR approach to improve the adolescents’ alcohol resistance ability, and to explore the user experiences for the VR FestLab.

However, in my opinion, I think the study need some more in-depth detailed comparison and quantitative description. And what’s more, whether this kind of VR technology is still effective for adolescents' alcohol resistance ability after long-term use, needs further verification and discussion.

Minor comments

This study has three main objectives, one is to improve the alcohol resistance ability among adolescents, and another is to explore their user experiences when using VR FestLab, and then the third one indicate that the VR FestLab is a promising tool for the prevention of alcohol use. I have some comments listed below, for reference and discussion:

  1. For sample selection, although the authors also mentioned that the students are come from different regions and have different cultural backgrounds, but I still think that the sample from one Danish boarding school may not be representative. Because for this huge group, it should include adolescents both in school and out of school, in boarding school and other forms of school, etc. The situation of adolescents is not the same in different growth environment, maybe the results will be different.
  2. For the improvement of alcohol resistance ability, perhaps it is not enough to emphasize subjectively whether the skill has improved or not. By adding quantitative evaluation index is an acceptable way. For instance, the alcohol resistance ability was quantified into a specific value A, and finally changed from value A to value B after using VR FestLab, compared with it's not been used.
  3. For the evaluation of the alcohol prevention method, as the authors mentioned in the study, the VR FestLab is a promising tool for the prevention of alcohol use. But I think its better to add a result comparison between this VR method and other traditional methods. By comparison, we can intuitively see that the VR FestLab is indeed more promising.
  4. For the common disadvantages, there are fewer cases recorded in this paper. May be due to the small number of samples or the short experimental period. But VR technology may cause vertigo, myopia, visual fatigue and other problems, and it may also have an opposite effect of making these adolescents addicted to other video games. As a promising prevention tool for alcohol education program, how does the VR FestLab overcome these problems?
  5. For the user experiences, I think it should include at least two parts, one is the subjective use feeling mentioned in the paper, for instance, the scenes was real or not, the gameplay design was user-friendly or not, etc. Another part should focus on the use effect of this VR product. I suggest adding several comparison groups, listed below for reference:
  • Comparison of drinking group and non drinking group. As the authors mentioned in the conclusion, further research may be needed. Because different people have different feelings about the products.
  • Comparison of short-term and long-term use. The feedback on the use of a same product maybe different when in short-term and long-term use.
  • Comparison between virtual simulation and real-life test. There are many complicated situations and scenes in the real world, not just typical party situation. I think the effect of VR training needs to be further tested.

Recommendation

I suggest adding some quantitative evaluation and comparison to verify the effectiveness of this VR products after long-term testing.

Round 2

Reviewer 2 Report

The authors have cleared all of my concerns. So i think it can be accepted now.